# Numerical Simulation of a Time-Dependent Electroviscous and Hybrid Nanofluid with Darcy-Forchheimer Effect between Squeezing Plates

**DOI:** 10.3390/nano12050876

**Published:** 2022-03-06

**Authors:** Muhammad Sohail Khan, Sun Mei, Unai Fernandez-Gamiz, Samad Noeiaghdam, Aamir Khan

**Affiliations:** 1School of Mathematical Sciences, Jiangsu University, Zhenjiang 212013, China; sohailkhan8688@gmail.com (M.S.K.); shabnam8688@gmail.com (S.); 2Nuclear Engineering and Fluid Mechanics Department, University of the Basque Country UPV/EHU, Nieves Cano 12, 01006 Vitoria-Gasteiz, Spain; unai.fernandez@ehu.eus; 3Department of Applied Mathematics and Programming, South Ural State University, Lenin Prospect 76, 454080 Chelyabinsk, Russia; noiagdams@susu.ru; 4Industrial Mathematics Laboratory, Baikal School of BRICS, Irkutsk National Research Technical University, 664074 Irkutsk, Russia; 5Department of Pure and Applied Mathematics, University of Haripur, Haripur 22620, Khyber Pakhtunkhwa, Pakistan; aamir.khan@uoh.edu.pk

**Keywords:** hybrid nanoliquids (MoS_2_ + Au/C_2_H_6_O_2_ − H_2_O), Darcy-Forchheimer flow, electric potential, electrioviscous effect, parametric continuation method (PCM), BV4C schemes and HAM

## Abstract

In this article, the behavior of transient electroviscous fluid flow is investigated through squeezing plates containing hybrid nanoparticles. A hybrid nanofluid ${\mathrm{MoS}}_{2}+\mathrm{Au}$/$C_{2}H_{6}O_{2}-H_{2}O$ was formulated by dissolving the components of an inorganic substance such as molybdenum disulfide $\left({\mathrm{MoS}}_{2}\right)$ and gold (Au) in a base fluid of ethylene glycol/water. This hybrid non-liquid flow was modeled by various nonlinear mathematical fluid flow models and subsequently solved by numerical as well as analytical methods. For the numerical solution of nonlinear ODEs, a built-in function BVP4C was used in MATLAB, and the same problem was solved in MATHEMATICA by HAM. The result of the present problem related to the results obtained from the existing literature under certain conditions. The outcomes revealed that the concentration profiles were more sensitive to homogeneity diversity parameters. The simulation of the various physical parameters of the model indicated that the heat transfer through a mixture of hybrid nanofluids was greater than a simple nanofluid. In addition, the phenomenon of mixed convection was considered to improve the velocity of simple nanofluids and hybrid nanofluids, when both cases have low permeability. A rise in the volume fraction of the nanomaterials, Φ, was associated with an increase in the heat transfer rate. It was observed that the heat transfer rate of the hybrid nanofluids ${\mathrm{MoS}}_{2}+{\mathrm{Au}/C}_{2}H_{6}O_{2}-H_{2}O$ was higher than that of the single nanofluids ${\mathrm{MoS}}_{2}/C_{2}H_{6}O_{2}-H_{2}O$.

## 1. Introduction

Our survival in this modern age depends heavily on machines and technology. We are updating them in many ways for efficient use. A few years ago, a new technology was introduced to immerse nanosized materials into base liquids to improve the cooling capacity of certain machines. Nanofluids are used to improve the thermal conductivity and heat transfer rate of the base fluids. Typically, the fundamental liquids are triethylene glycol, oil, water, polymeric solutions, refrigerants, bioliquids, ethylene, and lubricants. The most commonly used nanoparticles are copper, gold, zirconium, aluminum, metal nitride AIN, Al${}_{2}$O${}_{3}$, diamond, carbon nanotubes, CuO metal carbides, and SiC. There are lots of qualities associative with nanofluids such as uniformity, long-lasting stability, elevated thermal conductivity at low nanomaterials clustering, and minimal obstruction of flow paths. Due to above mentioned qualities this fluid built many applications in electrical devices such as micro-reactors and fluidic digital display, micro-electromechanical systems, etc. In addition, nanomaterials are the main constituent of food items, agriculture, medicines, and nuclear reactors. Nanofluids are used in various aspects of our lives such as thermal expansion and contraction of buildings, cancer therapeutics, heat interchanges, cryopreservation, imaging, refrigeration of electronic apparatus, pharmaceutical processes, the transportation industry, sensing, and microfluidics. The idea of an improved heat transfer fluid by suspending metallic nanomaterials in fundamental fluids was introduced by Choi and Eastman [1]. The newly updated fluid with improved thermal conductivity was called a nanofluid. The suspension of nanomaterials into a base fluid enhances the thermal conductivity of the base fluid, for example, water, engine oil, and ethylene glycol. An empirical investigation of the suspension of copper nanomaterials in ethylene glycol was conducted by Eastman et al. [2], concluded that the thermal conductivity of ethylene glycol increases up to 40 percent due to suspended nanomaterials, which increase the ability of heat transmission in the flow. Nanofluids have been used in a variety of applications, such as fuel reduction in electric power plants, vehicle cooling, and the delivery of nanodrugs [3]. Wong and De Leon [4] reported that the use of nanofluidic coolants in devices saves energy and reduces emissions, resulting in lower production costs. Later, Buongiorno [5] analyzed the relative velocity of nanomaterials and conventional fluids produced by the seven-slip mechanism and concluded that only thermophoresis and Brownian motion were responsible for improving heat transmission in the nanofluid. Compression of two plates under externally applied pressure produces a squeezing flow. The concept of squeezing flow is used in different engineering applications, for instance, driven pistons, lubricating processes, hydraulic lifts, and injection molding. Stefan [6] studied non-Newtonian fluid motion between two moving surfaces through lubricating processes. After this work, Stefan researched the squeezing flow and investigated its motion in various geometries. Subsequently, Archibald [7] and Reynolds [8] analyzed the behavior of squeezing flow in rectangular and elliptical geometries, and their governing equations were modeled through the Reynolds equation. Nevertheless, it is not appropriate to apply high velocity and porous thrust bearings to the squeezing flow, as described in Jackson [9] and Ishizawa [10]. The Jeffrey fluid depends on the applied shear stress, which is why it has been classified as non-Newtonian fluid. The liquid behaves as a solid if the applied shear stress is lower than the yield stress, while the liquid starts to flow if the applied shear stress is higher than the yield stress [11,12]. Jeffrey fluid’s mathematical formulation used time derivatives as convective derivatives. In the Jeffrey model, the retardation times and relaxation parameters are important for the illustration of viscoelastic characteristics in polymer sectors [13]. The Jeffrey fluid model is considered a model of blood flow to the arteries [14]. Many scientists have recognized the importance of MHD flow on the boundary layer, because this concept is used in various engineering applications. A Lorentz force can be produced by imposing a magnetic field on the electrically conducted fluid. Lorentz Force applications are found in various instruments such as mass spectrometers and cyclotrons [15,16]. Recently, advances in Darcy’s law have increased research into the fluid flow in a porous medium. The use of a porous medium in engine cooling devices is recommended, as it is useful to increase the heat dissipation in the devices [17]. Radhakrishnamacharya and Nallapu [18] studied the behavior of Jeffrey fluid under MHD flow in a permeable circular channel. Subsequently, Ahmad and Ishak [19] observed the mixed convection flow of Jeffrey fluid at the stagnation point on stretched vertical plates with a magnetic field. Microchannel flows are frequently used for the segregation of chemical and biological components, such as DNA segregation in the genetic field. This procedure is called electrophoresis [20,21]. It applies the principle of various mobilities of charged particles under an applied electric field to direct them in opposite directions. This type of induced flow is known as electroosmotic flow. The region through which the flow enters a microchannel is called the entry region. In this region, skin friction and velocity profiles show significant changes in the direction of the stream. In a fully developed region, the skin friction and velocity do not show fluctuation in the streamwise direction. Many factors influence the physics of microchannel flow, such as the electric field, the electrical charge density, and the zeta potential of the microchannel. These factors can be included in the classical Navier–Stokes equation for a microchannel flow. Yang et al. [22] and Arulanandam and Li [23] numerically solved the fluid flow model in the microchannel. Their model used the Poisson-Boltzmann equation to describe the behavior of the electrical double layer potential; the Laplace equation explained the electrostatic field, and the modified version of the Navier–Stokes equations described the influence of the body forces that occur as a result of the interaction between zeta and the electrical potential. The numerical outcomes of their model were in line with their empirical observations. In existing and new biotechnologies application, it is important to understand the flow in micrometer-sized channels to improve new microfluidic instruments with the power to manipulate and transport liquid on a small scale [24]. In the last few years, much research has been conducted in the field of microfluidic and microscale instruments [25]. Surface tension and electrokinetic influence play a significant role in surface phenomena at micrometer scales [26,27].

Khan et al. [28,29,30] studied nanofluid flow in terms of mass and heat transfer in the presence of variable magnetic fields. In their proposed model, they analyzed the behavior of (Cu-H${}_{2}$O) nanofluid between two parallel discs under the influence of a variable magnetic field. Khan et al. [31,32,33,34] reviewed the Poisson-Boltzmann formulation, which comes from the assumption of thermodynamic equilibrium by imposing the condition that the distribution of the ions would not be affected by the movement of the fluid. However, while this is a reasonable assumption for the steady flow of electroosmotic liquids via linear microchannels, there are some important cases where the convective transport of ions has considerable influence. In these cases, the Nernst–Planck model should be used instead of the Poisson-Boltzmann model to generate an electric field in the region under consideration. Sajid et al. [35] examined the incompressible Prandle number micropolar fluid motion at a porous stretching sheet along with the impact of a temperature-dependent exponential heat source, viscous dissipation, higher-order chemical reaction, convective boundary conditions, and nonlinear thermal radiation. Jamshed et al. [36] examined the transient flow behavior of the Casson non-Newtonian nanofluid with respect to entropy generation and thermal transport. The influences of thermal solar transport and slip conditions for the Casson nanofluid flow were thoroughly investigated. To analyze nanofluids regarding flow behavior and thermal transport, slippery surface conditions were imposed under convective heat. The mathematical formulations for the Casson nanofluid flow and heat transfer were summarized through presumed boundary conditions. Waqas et al. [37] observed that the increasing need for modern technology to better enhance the heat transfer efficiency of thermal systems has made nanofluids more important in the last two decades. Keeping such applications in mind, the presence of the Carreau-Yasuda nanofluid moving microorganisms, thermal radiation, and the conditions of Robin’s boundaries were examined.

Succeed in previous works [28,29], persuading to examine the joint effect of the electromagnetic force, inertia force, surface suction/injection, magnetic field, and ionized fluid in the squeezing flow of hybrid nanofluids between two parallel plates. We assumed that the lower plate was stretchable and permeable with a linear velocity. The mathematical model of $({\mathrm{MoS}}_{2})/\mathrm{Au}$ + base fluid ethylene glycol-water was formulated and then transformed into reduced differential equations via the similarity transformation. In the squeezing flow, the aforementioned combination of influences has not been studied before. After ionization, we studied the behaviour of the fluid as well. The PCM/BVP4C/HAM application was used to generate the results, which were validated based on the available numerical values from the previous works. This study is a novel and original consideration of an internal hybrid nanofluid flow. Numerical computation and graphical representation were completed for the Nussult number, velocity field, skin frictions, temperature, and ion distribution.

## 2. Formulation

Molybdenum disulfide $\left({\mathrm{MoS}}_{2}\right)$ and gold (Au) in water and ethylene glycol formation was considered to flow between two infinite parallel plates, as shown in Figure 1. The upper plate with velocity $\frac{dh}{dt}=-\frac{\alpha }{2}\sqrt{\frac{\nu_{bf}}{b(1-\alpha t)}}$ was moving toward the lower plate, while the upper plate was placed at $y=h\left(t\right)=\sqrt{\frac{\nu_{bf}\left(1-\alpha t\right)}{b}}$ from the lower plate. Therefore, the porous lower plate was included in the physical illustration for the possible fluid suction/injection with the wall mass velocity denoted as $V_{w}=-\frac{V_{0}}{1-\alpha t}$; $V_{0}$ > 0 for suction, $V_{0}$ < 0 for injection, and $V_{0}$ = 0 corresponds to an impermeable plate. A further assumption was that the lower and upper plates were maintained at fixed temperatures T1 and T2, respectively. Furthermore, the lower plate was stretchable with the linear velocity $u_{w}=-\frac{bx}{1-\alpha t}$; $t<\frac{1}{\alpha }$, while the inclusion of the time-dependent magnetic field was formulated as $H=\frac{B_{0}}{1-\alpha t}$. The fundamental governing equations are as follows [28,29,30]:

Continuity equation:(1)$$\frac{\partial u}{\partial x}+\frac{\partial v}{\partial y}=0;$$

Momentum equations with electroviscous and uniform magnetic effect [28,29]:(2)$$\begin{array}{cc} \frac{\partial U}{\partial t}+u\frac{\partial U}{\partial x}+v\frac{\partial U}{\partial y} & =\frac{\mu_{hnf}}{\rho_{hnf}}\frac{\partial^{2}U}{\partial y^{2}}-\frac{\sigma_{hnf}}{\rho_{hnf}}H^{2}U-\frac{\mu_{hnf}}{\rho_{hnf}}\frac{U}{K^{*}}-FrU^{2}\\ & -\frac{BK^{2}\mu_{hnf}}{\rho_{hnf}}\left(n^{+}-n^{-}\right)\frac{\partial W}{\partial x}; \end{array}$$

Poisson Equation [38,39]:(3)$$\frac{\partial^{2}W}{\partial x^{2}}+\frac{\partial^{2}W}{\partial y^{2}}=-\frac{1}{2}K^{2}\left(n^{+}-n^{-}\right);$$

Nernst–Planck Equations [38,39]:(4)$$\begin{array}{c} \frac{\partial n^{+}}{\partial t}+u\frac{\partial n^{+}}{\partial x}+v\frac{\partial n^{+}}{\partial y}=\frac{\mu_{hnf}}{\rho_{hnf}Sc}\left(\frac{\partial^{2}n^{+}}{\partial y^{2}}+\frac{\partial n^{+}}{\partial x}\frac{\partial W}{\partial x}+\frac{\partial n^{+}}{\partial y}\frac{\partial W}{\partial y}+n^{+}\frac{\partial^{2}W}{\partial y^{2}}\right) \end{array}$$
(5)$$\begin{array}{c} \frac{\partial n^{-}}{\partial t}+u\frac{\partial n^{-}}{\partial x}+v\frac{\partial n^{-}}{\partial y}=\frac{\mu_{hnf}}{\rho_{hnf}Sc}\left(\frac{\partial^{2}n^{-}}{\partial y^{2}}-\frac{\partial n^{-}}{\partial x}\frac{\partial W}{\partial x}-\frac{\partial n^{-}}{\partial y}\frac{\partial W}{\partial y}-n^{-}\frac{\partial^{2}W}{\partial y^{2}}\right); \end{array}$$

Energy Equation [28]:(6)$$\begin{array}{c} \frac{\partial T}{\partial t}+u\frac{\partial T}{\partial x}+v\frac{\partial T}{\partial y}=\frac{\kappa_{hnf}}{{\left(\rho C_{p}\right)}_{hnf}}\frac{\partial^{2}T}{\partial y^{2}}+\frac{Q_{0}}{{\left(\rho C_{p}\right)}_{hnf}}\left(T-T_{0}\right); \end{array}$$

Mass Transfer Equation [29]:(7)$$\begin{array}{c} \frac{\partial C}{\partial t}+u\frac{\partial C}{\partial x}+v\frac{\partial C}{\partial y}=D_{B}\frac{\partial^{2}C}{\partial y^{2}}+\frac{D_{T}}{T_{1}}\frac{\partial^{2}T}{\partial y^{2}}, \end{array}$$
where *U* = $\frac{\partial v}{\partial x}$ − $\frac{\partial u}{\partial y}$ are the associate conditions at the lower and upper plates [28]. Here, ${\left(\rho C_{p}\right)}_{hnf}$ is the heat capacity of the hybrid nanofluid, *T* is the temperature, *C* is the concentration, $n^{+}$ and $n^{-}$ are positive and negative ions, $K^{*}$ is the permeability of porous space, $Fr=\frac{C_{b}^{*}}{\sqrt{K^{*}}}$ is the non-uniform inertia coefficient of the porous medium [40], $C_{b}^{*}$ is the drag coefficient, $B_{0}$ is the strength of the magnetic field, *W* is the electrical potential of ions, $B=\frac{\rho k^{2}T^{2}\varepsilon_{0}\varepsilon }{2z^{2}e^{2}\mu^{2}}$ is fixed at a constant temperature, $Q_{0}$ is the heat generation, $K^{2}=\frac{2z^{2}e^{2}n_{0}}{\varepsilon_{0}\varepsilon k_{b}T}$ is the inverse Debye constant, $\rho_{hnf}$ is the fluid density of the hybrid nanofluid, $\sigma_{hnf}$ is electrical conductivity of hybrid nanofluid, and $\mu_{hnf}$ is the kinematic viscosity of the hybrid nanofluid.

Nanofluids are defined as [28]:(8)$$\begin{array}{cc} \; & \nu_{hnf}=\frac{\mu_{hnf}}{\rho_{hnf}},\;\frac{\rho_{hnf}}{\rho_{bf}}=\left(1-\left(\Phi_{1}+\Phi_{2}\right)\right)+\Phi_{1}\frac{\rho_{M}}{\rho_{bf}}+\Phi_{2}\frac{\rho_{2}}{\rho_{bf}},\\ & \frac{\kappa_{hnf}}{\kappa_{bf}}=\frac{\kappa_{1}+2\kappa_{bf}-2\Phi_{2}\left(\kappa_{bf}-\kappa_{2}\right)}{\kappa_{1}+\kappa_{bf}+\Phi_{2}\left(\kappa_{bf}-\kappa_{2}\right)}\\ & \;\frac{{\left(\rho Cp\right)}_{hnf}}{{\left(\rho C_{p}\right)}_{bf}}=\left(1-\Phi_{2}\right)\left(\left(1-\Phi_{1}+\Phi_{1}\frac{{\left(\rho C_{p}\right)}_{1}}{{\left(\rho C_{p}\right)}_{bf}}\right)+\Phi_{2}\frac{{\left(\rho C_{p}\right)}_{2}}{{\left(\rho C_{p}\right)}_{bf}}\right),\\ & \frac{\sigma_{hnf}}{\sigma_{bf}}=\frac{\sigma_{1}+2\sigma_{bf}-2\Phi_{2}\left(\sigma_{bf}-\sigma_{2}\right)}{\sigma_{1}+\sigma_{bf}+\Phi_{2}\left(\sigma_{bf}-\sigma_{2}\right)},\;\frac{\mu_{hnf}}{\mu_{bf}}=\frac{1}{{\left(1-\left(\Phi_{1}+\Phi_{2}\right)\right)}^{2.5}}, \end{array}$$
where $\kappa_{hnf}$ is the thermal conductivity of the hybrid nanofluid, $\kappa_{bf}$ is the thermal conductivity of the base fluid, and $\Phi_{1}$ and $\Phi_{2}$ are the volume fractions of hybrid nanoparticles.

The boundary conditions of the proposed model are as follows [28]:(9)$$\begin{array}{cc} \; & u=\lambda \frac{bx}{1-\alpha t},\;v=-\frac{V_{0}}{1-\alpha t},\;T=T_{1},\;C=C_{1},\\ & \;W=0,\;n^{+}=n^{-}=0,\;\;at\;y=0\\ & u=0,\;v=\frac{dh(t)}{dt},\;T=T_{2},\;C=C_{2},\;\\ & W=\frac{x}{l^{2}\left(1-\alpha t\right)},\;n^{+}=n^{-}=\frac{\alpha }{\nu_{bf}\left(1-\alpha t\right)},\;\;\;at\;y=h\left(t\right). \end{array}$$

The following similarity transformations [28] are considered for reducing the PDE (Equation 1)–(Equation 7) to an ODE system,
(10)$$\begin{array}{cc} \; & \psi =\sqrt{\frac{b\nu_{bf}}{1-\alpha t}}xf\left(\eta \right),\;u=\frac{bx}{1-\alpha t}f^{\prime }\left(\eta \right),\;v=-\sqrt{\frac{b\nu_{bf}}{1-\alpha t}}f\left(\eta \right),\;\\ & \theta =\frac{T-T_{1}}{T_{2}-T_{1}},\;\phi =\frac{C-C_{1}}{C_{2}-C_{1}},\;W=\frac{x}{l^{2}\left(1-\alpha t\right)}P\left(\eta \right),\;\\ & n^{+}=\frac{\alpha }{\nu_{bf}\left(1-\alpha t\right)}G\left(\eta \right),\;n^{-}=\frac{\alpha }{\nu_{bf}\left(1-\alpha t\right)}H\left(\eta \right),\;\eta =y\sqrt{\frac{b}{\nu_{bf}\left(1-\alpha t\right)}}. \end{array}$$

Therefore, Equation (Equation 1) is satisfied and the remaining Equations (Equation 2)–(Equation 7) transform into the following form
(11)$$\begin{array}{cc} f^{⁗} & =\frac{\rho_{hnf}/\rho_{bf}}{\mu_{hnf}/\mu_{bf}}\left(\frac{Sq}{2}\left(\eta f^{\prime \prime \prime }+3f^{\prime }\right)-ff^{\prime \prime \prime }+f^{\prime }f^{\prime \prime }-Fr{f^{\prime \prime }}^{2}\right)\\ & -\frac{\sigma_{hnf}/\sigma_{bf}}{\mu_{hnf}/\mu_{bf}}Mf^{\prime \prime }+K_{1}^{*}f^{\prime \prime }+BK^{2}RH\left(G-H\right), \end{array}$$
(12)$$P^{\prime \prime }=-\frac{1}{2}K^{2}\delta_{1}\left(G-H\right),$$
(13)$$G^{\prime \prime }=\frac{Sc\rho_{hnf}/\rho_{bf}}{\mu_{hnf}/\mu_{bf}}\left(\frac{Sq}{2}\left(2G+\eta G^{\prime }\right)-fG^{\prime }\right)-\frac{1}{\delta_{1}}\left(G^{\prime }P^{\prime }-\frac{K^{2}\delta_{1}}{2}\left(G^{2}-GH\right)\right),$$
(14)$$H^{\prime \prime }=\frac{Sc\rho_{hnf}/\rho_{bf}}{\mu_{hnf}/\mu_{bf}}\left(\frac{Sq}{2}\left(2H+\eta H^{\prime }\right)-fH^{\prime }\right)-\frac{1}{\delta_{1}}\left(H^{\prime }P^{\prime }-\frac{K^{2}\delta_{1}}{2}\left(GH-H^{2}\right)\right),$$
(15)$$\theta^{\prime \prime }=\frac{{\left(\rho Cp\right)}_{hnf}/{\left(\rho Cp\right)}_{bf}}{\kappa_{hnf}/\kappa_{bf}}\left(\frac{SqPr}{2}\eta \theta^{\prime }-Prf\theta^{\prime }-\frac{Pr}{{\left(\rho Cp\right)}_{hnf}/{\left(\rho Cp\right)}_{bf}}Q\theta \right),$$
(16)$$\begin{array}{cc} \phi^{\prime \prime } & =\frac{SqLe}{2}\eta \phi^{\prime }-Lef\phi^{\prime }-\frac{Nt}{Nb}\frac{{\left(\rho Cp\right)}_{hnf}/{\left(\rho Cp\right)}_{bf}}{\kappa_{hnf}/\kappa_{bf}}(\frac{SqPr}{2}\eta \theta^{\prime }\\ & -Prf\theta^{\prime }-\frac{Pr}{{\left(\rho Cp\right)}_{hnf}/{\left(\rho Cp\right)}_{bf}}Q\theta ), \end{array}$$
where $M=\frac{\sigma_{bf}B_{0}^{2}}{b\rho_{bf}}$ is the magnetic parameter, $Pr=\frac{\mu_{bf}Cp_{bf}}{\kappa_{bf}}$ is the Prandtl number, $Sq=\frac{\alpha }{b}$ is the squeeze parameter, $K_{1}^{*}=\frac{\nu_{bf}\left(1-\alpha t\right)}{K^{*}b}$ is the local porosity parameter, λ>0 refers to the stretching lower plate, λ=0 denotes the fixed/static lower plate, Fr is the Forchheimer number [40], $Sc=\frac{\mu_{bf}}{\rho_{bf}D}$ is the Schmidt number, the thermophoresis parameters are $Nt=\frac{D_{T}\left(T_{2}-T_{1}\right)}{T_{1}\nu_{bf}}$, $S=\frac{V_{0}}{lb}$ is the suction/injection parameter, $Le=\frac{\nu_{bf}}{D_{B}}$ is the Lewis number, $\delta_{1}=\frac{\alpha^{2}}{l^{2}}$, $Q=\frac{Q_{0}}{b{\left(\rho Cp\right)}_{bf}}$ is the heat source/sink substrate, and the Brownian motion is $Nb=\frac{D_{B}\left(C_{2}-C_{1}\right)}{\nu_{bf}}$. $T_{0}$ and $C_{0}$ are the ambient temperature and nanoparticle concentration, $T_{2}$ and $C_{2}$ are any reference temperature and concentration chosen unequal to $T_{0}$ and $C_{0}$ [28,40], with constants $\delta =\frac{T_{1}-T_{0}}{T_{2}-T_{0}}$ and $\omega =\frac{C_{1}-C_{0}}{C_{2}-C_{0}}$.

The boundary conditions in their reduced form are as follows,
(17)$$\begin{array}{cc} \; & f^{\prime }\left(0\right)=\lambda ,\;f\left(0\right)=S,\;\theta \left(0\right)=\delta ,\;\phi \left(0\right)=\omega ,\;\\ & P(0)=0,\;G(0)=0,\;H(0)=0,\;\;at\;y=0\\ & f^{\prime }\left(1\right)=0,\;f\left(1\right)=\frac{sq}{2},\;\theta \left(1\right)=1,\;\phi \left(1\right)=1,\;\\ & P(1)=1,\;G(1)=1,\;H(1)=1,\;\;at\;y=1. \end{array}$$

The required physical parameters are the Nusselt number and the skin friction coefficient at the upper plate and lower plate, which can be defined as,
(18)$${\left(Re_{x}\right)}^{\frac{1}{2}}C_{f_{u}per}=\frac{\mu_{hnf}}{\mu_{bf}}f^{\prime \prime }\left(1\right),\;\;\;\;{\left(Re_{x}\right)}^{\frac{1}{2}}C_{f_{l}ower}=\frac{\mu_{hnf}}{\mu_{bf}}f^{\prime \prime }\left(0\right),$$
and
(19)$${\left(Re_{x}\right)}^{\frac{-1}{2}}N_{u_{u}per}=-\frac{\kappa_{hnf}}{\kappa_{bf}}\theta^{\prime }\left(1\right),\;\;\;\;{\left(Re_{x}\right)}^{\frac{-1}{2}}N_{u_{l}ower}=-\frac{\kappa_{hnf}}{\kappa_{bf}}\theta^{\prime }\left(0\right),$$
where $Re_{x}=\frac{xU_{w}}{\nu_{bf}}$.

## 3. Numerical Solution by PCM

This section presents the numerical solution of the proposed mathematical model for the best selection of the continuation parameters through the algorithm of a well-known numerical scheme (PCM). The PCM algorithm is used to solve the nonlinear ODEs in (Equation 11)–(Equation 16) through the boundary condition given in Equation (Equation 17):**First order of ODE**We use the variables below to transform the PDEs given in (11)–(16) into the first order of ODEs.
(20)$$\begin{array}{cc} \; & f=P_{1},\;f^{\prime }=P_{2},\;f^{\prime \prime }=P_{3},\;f^{\prime \prime \prime }=P_{4},\;P=P_{5},\;P^{\prime }=P_{6},\\ & G=P_{7},\;G^{\prime }=P_{8},\;H=P_{9},\;H^{\prime }=P_{10},\;\\ & \theta =P_{11},\;\theta^{\prime }=P_{12},\;\phi =P_{13},\;\phi^{\prime }=P_{14} \end{array}$$By putting these new variables into Equations (Equation 11)–(Equation 16), we obtain the following transformed equations as below.
(21)$$\begin{array}{cc} P_{4}^{\prime } & =\frac{\rho_{nf}/\rho_{bf}}{\mu_{nf}/\mu_{bf}}\left(\frac{Sq}{2}\left(\eta P_{4}+3P_{2}\right)-P_{1}P_{4}+P_{2}P_{3}-FrP_{3}^{2}\right)\\ & -\frac{\sigma_{nf}/\sigma_{bf}}{\mu_{nf}/\mu_{bf}}MP_{3}+K_{1}^{*}P_{3}+BK^{2}RH\left(P_{7}-P_{9}\right), \end{array}$$
(22)$$P_{6}^{\prime }=-\frac{1}{2}K^{2}\delta_{1}\left(P_{7}-P_{9}\right),$$
(23)$$P_{8}^{\prime }=\frac{Sc\rho_{nf}/\rho_{bf}}{\mu_{nf}/\mu_{bf}}\left(\frac{Sq}{2}\left(2P_{7}+\eta P_{8}\right)-P_{1}P_{8}\right)-\frac{1}{\delta_{1}}\left(P_{6}P_{8}-\frac{K^{2}\delta_{1}}{2}\left(P_{7}^{2}-P_{7}P_{9}\right)\right),$$
(24)$$P_{10}^{\prime }=\frac{Sc\rho_{nf}/\rho_{bf}}{\mu_{nf}/\mu_{bf}}\left(\frac{Sq}{2}\left(2P_{9}+\eta P_{10}\right)-P_{1}P_{10}\right)-\frac{1}{\delta_{1}}\left(P_{6}P_{10}-\frac{K^{2}\delta_{1}}{2}\left(P_{7}P_{9}-P_{9}^{2}\right)\right),$$
(25)$$P_{12}^{\prime }=\frac{{\left(\rho Cp\right)}_{nf}/{\left(\rho Cp\right)}_{bf}}{\kappa_{nf}/\kappa_{bf}}\left(\frac{SqPr}{2}\eta P_{12}-PrP_{1}P_{12}-\frac{Pr}{{\left(\rho Cp\right)}_{nf}/{\left(\rho Cp\right)}_{bf}}QP_{11}\right),$$
(26)$$\begin{array}{cc} P_{14}^{\prime } & =\frac{SqLe}{2}\eta P_{14}-LeP_{1}P_{14}-\frac{Nt}{Nb}\frac{{\left(\rho Cp\right)}_{nf}/{\left(\rho Cp\right)}_{bf}}{\kappa_{nf}/\kappa_{bf}}(\frac{SqPr}{2}\eta P_{12}\\ & -PrfP_{12}-\frac{Pr}{{\left(\rho Cp\right)}_{nf}/{\left(\rho Cp\right)}_{bf}}QP_{11}), \end{array}$$
and the boundary conditions become
(27)$$\begin{array}{cc} \; & P_{2}\left(0\right)=\lambda ,\;P_{1}\left(0\right)=S,\;P_{5}\left(0\right)=0,\;P_{7}\left(0\right)=0,\;P_{9}\left(0\right)=0,\;\\ & P_{11}\left(0\right)=\delta ,\;P_{13}\left(0\right)=\omega ,\;\;at\;y=0\\ & P_{2}^{\prime }\left(1\right)=0,\;P_{1}\left(1\right)=\frac{sq}{2},\;P_{5}\left(1\right)=1,\;P_{7}\left(1\right)=1,\;P_{9}\left(1\right)=1,\;\\ & P_{11}\left(1\right)=1,\;P_{13}\left(1\right)=1,\;\;at\;y=1 \end{array}$$**Introducing parameter *q* gives ODEs in the *q*-parameter group**To obtain ODEs in the *q*-parameter group, we need the *q*-parameter in Equations (Equation 21)–(Equation 26); therefore,
(28)$$\begin{array}{cc} P_{4}^{\prime } & =\frac{\rho_{nf}/\rho_{bf}}{\mu_{nf}/\mu_{bf}}\left(\frac{Sq}{2}\left(\eta \left(P_{4}-1\right)q+3P_{2}\right)-P_{1}\left(P_{4}-1\right)q\right)+P_{2}P_{3}-FrP_{3}^{2}\\ & -\frac{\sigma_{nf}/\sigma_{bf}}{\mu_{nf}/\mu_{bf}}MP_{3}+K_{1}^{*}P_{3}+BK^{2}RH\left(P_{7}-P_{9}\right), \end{array}$$
(29)$$P_{6}^{\prime }=-\frac{1}{2}K^{2}\delta_{1}\left(P_{7}-P_{9}\right),$$
(30)$$P_{8}^{\prime }=\frac{Sc\rho_{nf}/\rho_{bf}}{\mu_{nf}/\mu_{bf}}(\frac{Sq}{2}\left(2P_{7}+\eta \left(P_{8}-1\right)q\right)-P_{1}\left(P_{8}-1\right)q)-\frac{1}{\delta_{1}}(P_{6}\left(P_{8}-1\right)q)-\frac{K^{2}\delta_{1}}{2}(P_{7}^{2}-P_{7}P_{9}),$$
(31)$$P_{10}^{\prime }=\frac{Sc\rho_{nf}/\rho_{bf}}{\mu_{nf}/\mu_{bf}}(\frac{Sq}{2}\left(2P_{9}+\eta \left(P_{10}-1\right)q\right)-P_{1}\left(P_{10}-1\right)q)-\frac{1}{\delta_{1}}(P_{6}\left(P_{10}-1\right)q)-\frac{K^{2}\delta_{1}}{2}(P_{7}P_{9}-P_{9}^{2}),$$
(32)$$P_{12}^{\prime }=\frac{{\left(\rho Cp\right)}_{nf}/{\left(\rho Cp\right)}_{bf}}{\kappa_{nf}/\kappa_{bf}}\left(\frac{SqPr}{2}\eta \left(P_{12}-1\right)q\right)-PrP_{1}\left(P_{12}-1\right)q-\frac{Pr}{{\left(\rho Cp\right)}_{nf}/{\left(\rho Cp\right)}_{bf}}QP_{11},$$
(33)$$\begin{array}{cc} P_{14}^{\prime } & =\frac{SqLe}{2}\eta \left(P_{14}-1\right)q-LeP_{1}\left(P_{14}-1\right)q-\frac{Nt}{Nb}\frac{{\left(\rho Cp\right)}_{nf}/{\left(\rho Cp\right)}_{bf}}{\kappa_{nf}/\kappa_{bf}}(\frac{SqPr}{2}\eta P_{12}\\ & -PrfP_{12}-\frac{Pr}{{\left(\rho Cp\right)}_{nf}/{\left(\rho Cp\right)}_{bf}}QP_{11}), \end{array}$$**Differentiation by *q* reaches the following system with respect to the sensitivities of the parameter-*q***Differentiating the Equations (Equation 28)–(Equation 33) w.r.t by *q*
(34)$$d_{1}^{\prime }=h_{1}d_{1}+e_{1}$$
where $h_{1}$ is the coefficient matrix, $e_{1}$ is the remainder, and $d_{1}=\frac{dP_{i}}{d\tau }$, 1≤i≤14.**Cauchy Problem**(35)$$d_{1}=y_{1}+a1v_{1},$$
where $y_{1}$ and $v_{1}$ are the vector functions. By resolving the two types of Cauchy problems for each component, the system of the ODEs are satisfied automatically.
(36)$$e_{1}+h_{1}\left(a1v_{1}+y_{1}\right)={\left(a1v_{1}+y_{1}\right)}^{{}^{\prime }}$$
and we have the boundary conditions.**Use of Numerical Solution**An absolute scheme has been considered for the resolution of the problem
(37)$$\frac{v_{1}^{i+1}-v_{1}^{i}}{▵\eta }=h_{1}v_{1}^{i+1}$$
(38)$$\frac{y^{i+1}-y^{i}}{▵\eta }=h_{1}y^{i+1}+e_{1}$$**Take the corresponding coefficients**As mentioned, the boundaries are commonly used for $P_{i}$, where 1≤i≤14; for the solution of the ODEs, we used the equation $d_{2}=0$, and its corresponding matrix representation as given below.
(39)$$l_{1}.d_{1}=0\;\mathrm{or}\;l_{1}.\left(a1v_{1}+y_{1}\right)=0$$
where $a1=\frac{-l_{1}.y_{1}}{l_{1}.v_{1}}.$

## 4. Results and Discussion

In this section, the effects of different involved flow parameters on the velocity and electric field components are discussed graphically. The flow of hybrid nanofluids, variable electric fields, and heat/mass transfer phenomenon between two long parallel plates were observed. The system of highly nonlinear ODEs in Equations (Equation 11)–(Equation 16) with boundary conditions given in Equation (Equation 17) was solved analytically as well as numerically through HAM (Mathematica), PCM (Matlab), and BVP4C (Matlab). The local heat and mass transfer rate and the local skin friction coefficient on the surface of the squeezing plates were estimated, which was necessary in terms of the physical properties. The ongoing problem has a variety of physical parameters that plays a key role in the flow of fluid. The velocity profile, pressure distribution, mass, and heat transfers were calculated by solving the system of the ODEs in Equations (Equation 11)–(Equation 16), and their numerical results are graphically shown in Figure 2, Figure 3, Figure 4, Figure 5, Figure 6, Figure 7, Figure 8, Figure 9, Figure 10 and Figure 11 for various physical parameter values. In this section, the effects of the different flow parameters of the nonlinear ODEs in Equations (Equation 11)–(Equation 16) following the boundary conditions in Equation (Equation 17) are given. The statistics in Table 1 provide complete information about the thermophysical properties of the nanomaterials (Molybdenum disulfide (MoS_2_), Gold(Au)) and Ethylene-glycol with water ($C_{2}H_{6}O_{2}-H_{2}O$). It is necessary to mention that our proposed model produces good results when compared with the results of the models available in the current literature, and the comparison of the numerical results with (Najiyah et al. [28]) is shown in Table 2 and Table 3 for the upper and lower plates. Table 4 illustrates that the numerical and analytical outcomes of three important flow parameters, skin friction, and Nusselt number, which were obtained through three different numerical and analytical schemes (BVP4c, PCM, and HAM). The impacts of these fluid flow parameters are illustrated quantitatively through various tables and graphs for the velocity components $f^{\prime }\left(\eta \right)$, f(η), electrical potential P(η), electric field components G(η), H(η), temperature profile θ(η), and mass transfer. This section deals to analyze the influence of the involved parameters. It is necessary to note that the positive values of Sq reflect the top plate apart from the bottom plate whereas the negative values signify that the top plate attracted towards stationary bottom plate. The low or high value of Sq can be thought of as slow or rapid vertical velocity at the top plate, or the increment in the distance between the two plates, respectively.

Figure 2, Figure 3, Figure 4, Figure 5, Figure 6, Figure 7, Figure 8 and Figure 9 illustrate the behavior of the time-dependent squeezing flow in the presence of a magnetic field with cross-diffusion in two parallel disks. The impact of the suction/injection parameter (S) is depicted in Figure 2a,b. As the suction parameter increased (*S* = 1, 2, 3, 4), the velocity profile $f^{\prime }\left(\eta \right)$ decreased in Figure 2a, which suggests that a higher magnitude of suction strength may reduce the velocity distribution. The effect of the increase in injection parameter (*S* = −1, −2, −3, −4) causes the velocity profile to increase, as shown in Figure 2b. The impact of the squeezing number Sq on the transverse velocity at cold and hot surfaces for f(η) is shown graphically in Figure 3a. Influence of Sq at f(η) has been displayed graphically in both cases of ${\mathrm{MoS}}_{2}+{\mathrm{Au}/C}_{2}H_{6}O_{2}-H_{2}O$ and ${\mathrm{Au}/C}_{2}H_{6}O_{2}-H_{2}O$. In reality, the higher value of Sq causes the top plate to move down, which exerts an additional force on the nanomaterials; hence, the transverse velocity f(η) is improved. The effect of an increase in the inertia coefficient Fr on transverse velocity is seen in Figure 3b. According to this diagram, an increase in Fr causes the formation of a resistive force in fluid motion, resulting in a decrease in transverse velocity. Also, Figure 4a is represented to explain the situation when the dilating action of the walls occurs along with suction *S*. The channel’s subjacent area shows an increase in velocity of the fluid with the increasing magnetic parameter *M* likely because of the porosity of the walls suction *S*. Accordingly, the term S=2 suggests that the suction regulates the flow activity in the channel’s top portion and that increased fluid flow was observed. It was clearly shown from Figure 4a that the velocity of (${\mathrm{Au}/C}_{2}H_{6}O_{2}-H_{2}O$) nanofluid in the channel’s top portion is dominant as compared to the (${\mathrm{MoS}}_{2}+{\mathrm{Au}/C}_{2}H_{6}O_{2}-H_{2}O$) hybrid nanofluid. Gold (Au) also hold superiority in comparison with $MoS_{2}$, just in the channel’s top portion. The fluctuations Fr versus primary $f^{\prime }\left(\eta \right)$ secondary velocity distributions for hybrid nanofluid are highlighted in Figure 4b. In essence, a rise in Fr causes fluids to become more resilient, leading to decreased $f^{\prime }\left(\eta \right)$. Physically, boosting the amplitude of Fr reduces the interior nanofluid velocity. But after the middle area, the effect upon velocity distribution for rises of Fr. There is a reduction in velocity distribution, but the velocity distribution of the fluid flow increases when η>0.5. An increment in Sq leads to a reduction in the concentration profiles. Therefore, H(η) and G(η) increase and become a parabolic shape in Figure 5.

Figure 6 illustrates that the impact of H(η) and G(η) profile enhance due to the augmenting value of Schmidt number Sc for ${\mathrm{MoS}}_{2}+{\mathrm{Au}/C}_{2}H_{6}O_{2}-H_{2}O$ and ${\mathrm{Au}/C}_{2}H_{6}O_{2}-H_{2}O$ and so, the width of the concentration boundary layer is reduced. The reactant concentration rises rapidly as the diffusion coefficient of the species reduces, i.e., larger values of Sc cause a rapid increment in the concentration of the flow field, However, with increasing Schmidt number Sc values, the profile of H(η) reduces. The impact of the injection parameter *S* at θ(η) is explained for the higher absolute value of *S* in Figure 7a, which reflects that the upper plate moving down led to a decrease in the atomic collision in the nanocomponents and, consequently, a decline in the temperature and concentration profiles due to the injection parameter *S*, as shown in Figure 7b. The impact of *Q* on ϕ(η) and θ(η) can be seen in Figure 8 for both ${\mathrm{MoS}}_{2}+{\mathrm{Au}/C}_{2}H_{6}O_{2}-H_{2}O$ and ${\mathrm{Au}/C}_{2}H_{6}O_{2}-H_{2}O$. It has been concluded that the influence of *Q* (heat generation parameter) at the concentration profile θ(η) and ϕ(η) having an augmenting behaviour of velocity and temperature for ${\mathrm{MoS}}_{2}+{\mathrm{Au}/C}_{2}H_{6}O_{2}-H_{2}O$ and ${\mathrm{Au}/C}_{2}H_{6}O_{2}-H_{2}O$. The augmenting values of *Q* (heat generation parameter) reflect T2 > Tl. i.e., which signify more transfer of heat from surface to the liquid and consequently the temperature of the fluid is raised for both ${\mathrm{MoS}}_{2}+{\mathrm{Au}/C}_{2}H_{6}O_{2}-H_{2}O$ and ${\mathrm{Au}/C}_{2}H_{6}O_{2}-H_{2}O$. We observed an increment in the temperature profile with an increment in *Q*. Accordingly, there is an increase in uniform chemical reactions, which reduces viscosity. As can be seen in Figure 8b, the two properties of the fluid flow vary in opposite directions. In addition, reverse behavior is also seen for θ(η) in Figure 8a. The outcome is based on the assumption that heat generation would rise the nuclear energy in the fluid. Figure 9a,b depicts the effect of thermophoresis parameters and Brownian motion Nt and Nb on the concentration and temperature profile. The temperature distribution has been shown to increase monotonically from lower plate to the upper plate. It is worth noticing that the two key parameters Nt and Nb emphasis the thermophoresis and Brownian motion effects respectively. Figure 10 shows how the volume percentage of nanomaterials Φ affects the concentration profile ϕ(η) in both the (SNF) and (SNF) scenarios (HNF). It is clear that increasing the volume fraction of the nano-components Φ increases the velocity profile for both (HNF) and (SNF). The increase in the volume fraction of nanocomponents is owing to an indirect relationship between dynamic viscosity and the volume fraction of nanofluids, as demonstrated by the physical phenomena. As a result, the viscosity of the conventional fluid decreases in comparison to a rise in Φ, and the fluid flow increases. The effect of Le on ϕ is depicted in Figure 11. With an increase in Le, the volume percentage of nanoparticles increases, as does the thickness of the boundary layer. As a result, the concentration profile increases.

## 5. Concluding Remarks

The behaviour of a squeezing flow of hybrid nanoparticles under the influence of an electric field is explained in this article. BVP4C, PCM, and HAM were used to solve the suggested model of nanofluid flow numerically and analytically. The model’s solution was used to investigate the effects of physical parameters on the flow field, heat and mass profiles. The numerical outcomes of the proposed model were also validated using the results from the literature, and a good correlation was established. The following are the closing statements, which are based on the findings:The velocity profile shows gradually opposite behavior as a result of the augmenting value of the suction/injection parameter *S*.The increment in nanomaterial volume fraction reflects the same rising influence at the velocity profile for the value of *M* and *Sq*.The temperature profile and concentration profile declines with rising absolute values of nanomaterial volume fraction for S<0.With increasing values of nanomaterial volume fraction, the concentration profile rises.The higher the *M* and *S* values, the less skin friction there is at the wall.Skin friction is lower in the $\left({\mathrm{MoS}}_{2}\right)$ nanoparticles than in other ethylene glycol + water-based nanomaterials. Otherwise, $\left({\mathrm{MoS}}_{2}\right)$ heats the surface more efficiently than the rest of the mixtures.

## Figures and Tables

**Figure 1 nanomaterials-12-00876-f001:**
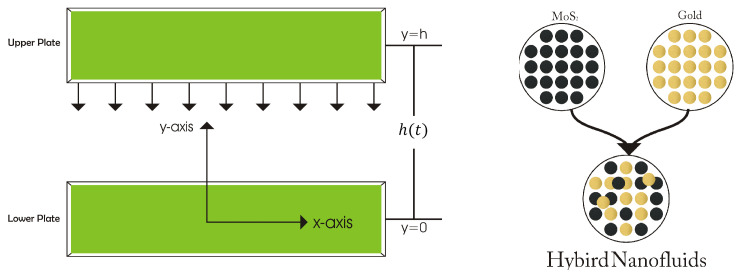
Geometry.

**Figure 2 nanomaterials-12-00876-f002:**
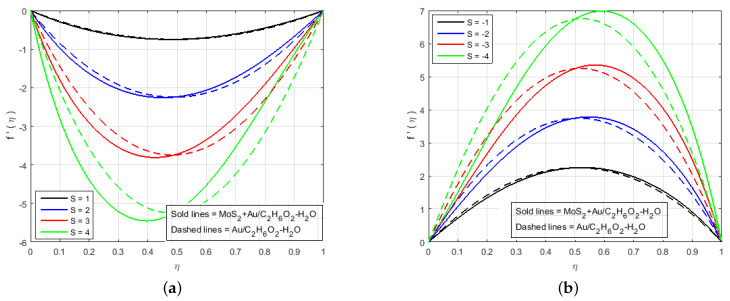
Impact of $f^{\prime }\left(\eta \right)$ for (**a**) S<0, (**b**) S>0 and fixed values of Sq = 1.0, *M* = 3.0, *B* = 0.6, *K* = 0.4, *Q* = 0.6, $K_{1}^{*}$ = 1.3, Le = 0.5, Sc = 0.7, Pr = 6.2, Nt = 0.4, Nb = 0.5, Fr = 1.5, $\Phi_{1}$ = 0.4, $\Phi_{2}$ = 0.04.

**Figure 3 nanomaterials-12-00876-f003:**
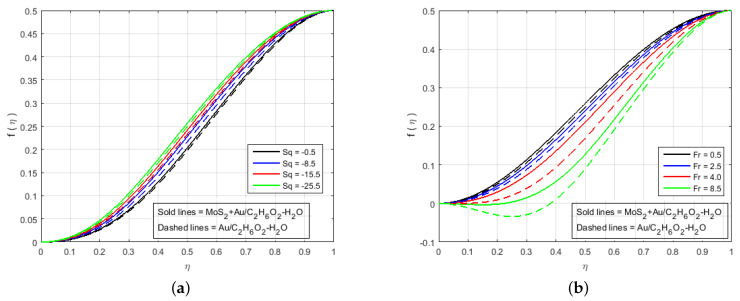
Impact of f(η) for (**a**) Sq and (**b**) Fr and fixed values of M=5.0, B=0.6, K=0.4, Le=0.5, Q=0.6, $K_{1}^{*}=1.3,\;$Sc=0.7, Pr=6.2, Nt=0.4, Nb=0.5, $\Phi_{1}=0.4,\;$$\Phi_{2}=0.04$.

**Figure 4 nanomaterials-12-00876-f004:**
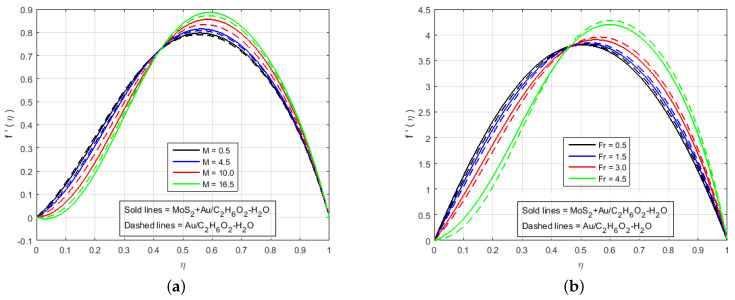
Impact of $f^{\prime }\left(\eta \right)$ for (**a**) M(Fr=1.5) and (**b**) Fr(M=3.0) and fixed values of Sq=−10.0, S=2.0, B=0.6, K=0.4, $K_{1}^{*}=1.3,\;$Le=0.5, Q=0.6, Sc=0.7, Pr=6.2, Nt=0.4, Nb=0.5, $\Phi_{1}=0.4,\;$$\Phi_{2}=0.04$.

**Figure 5 nanomaterials-12-00876-f005:**
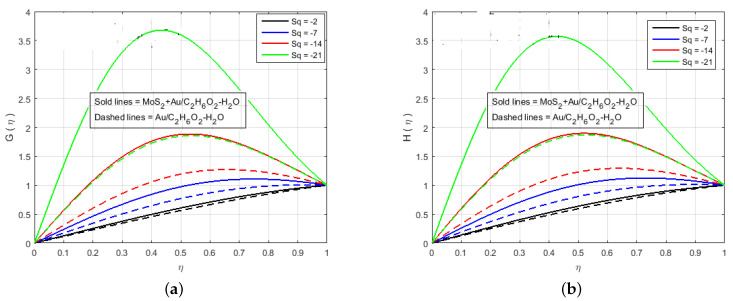
Impact of (**a**) G(η) and (**b**) H(η) for S<0 and fixed values of M=3.0, B=10.0, K=2.4, Le=0.5, Q=0.6, Sc=0.7, $K_{1}^{*}=1.3,\;$Pr=6.2, Nt=0.4, Nb=0.5, Fr=1.5, $\Phi_{1}=0.4,\;$$\Phi_{2}=0.04$.

**Figure 6 nanomaterials-12-00876-f006:**
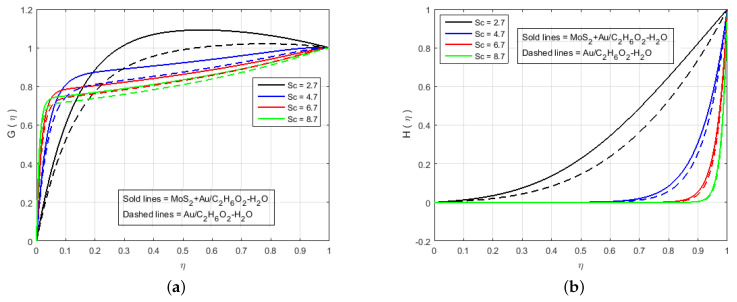
Impact of (**a**) G(η) and (**b**) H(η) for Sc and fixed values of M=3.0, B=10.0, K=2.4, Le=0.5, Q=0.6, $K_{1}^{*}=1.3,\;$S=−2.0, Pr=6.2, Nt=0.4, Nb=0.5, Fr=1.5, $\Phi_{1}=0.4,\;$$\Phi_{2}=0.04$.

**Figure 7 nanomaterials-12-00876-f007:**
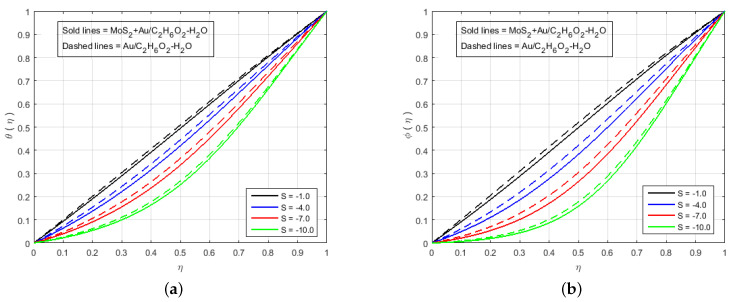
Impact of (**a**) θ(η) and (**b**) ϕ(η) for S<0 and fixed values of Sq=−1.0, M=5.0, B=0.6, K=1.4, $K_{1}^{*}=1.3,\;$Le=0.5, Q=0.6, Sc=0.7, Pr=6.2, Nt=1.4, Nb=1.5, Fr=1.5, $\Phi_{1}=0.4,\;$$\Phi_{2}=0.04$.

**Figure 8 nanomaterials-12-00876-f008:**
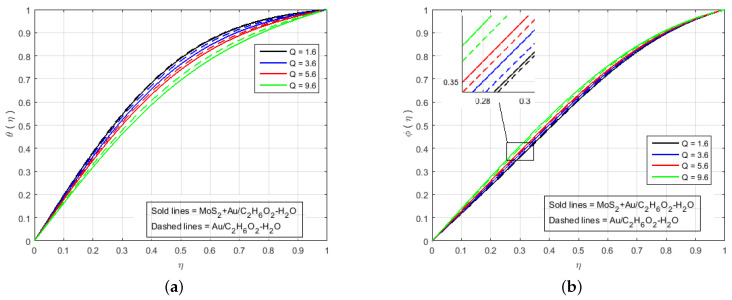
Impact of (**a**) θ(η) and (**b**) ϕ(η) for *Q* and fixed values of M=3.0, B=10.0, K=2.4, Le=0.5, $K_{1}^{*}=1.3,\;$S=10.0, Sc=0.7, Pr=6.2, Nt=1.4, Nb=1.5, Fr=1.5, $\Phi_{1}=0.4,\;$$\Phi_{2}=0.04$.

**Figure 9 nanomaterials-12-00876-f009:**
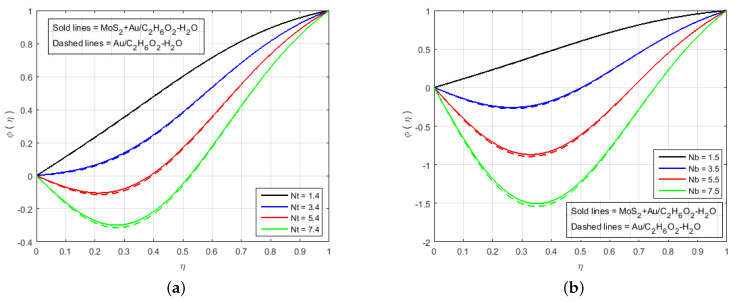
Impact of ϕ(η) for (**a**) Nt, (**b**) Nb and fixed values of M=3.0, B=10.0, K=2.4, Q=0.6, Le=0.5, Sc=0.7, $K_{1}^{*}=1.3,\;$Pr=6.2, S=−10.0, Fr=1.5, $\Phi_{1}=0.4,\;$$\Phi_{2}=0.04$.

**Figure 10 nanomaterials-12-00876-f010:**
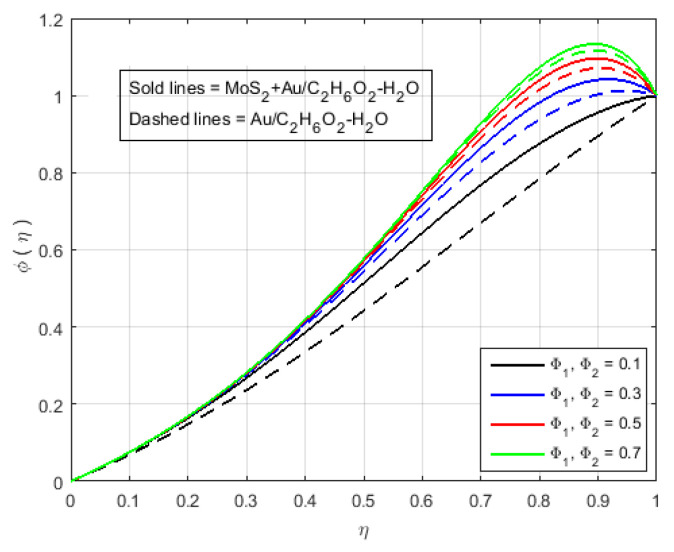
Impact of ϕ(η) for $\Phi_{1}$, $\Phi_{1}$ and fixed values of Sq=−5.0, M=3.0, B=10.0, K=2.4, Le=0.5, $K_{1}^{*}=1.3,\;$Q=0.6, Sc=0.7, Pr=6.2, S=−4.0, Fr=1.5, Nt=1.4, Nb=1.5.

**Figure 11 nanomaterials-12-00876-f011:**
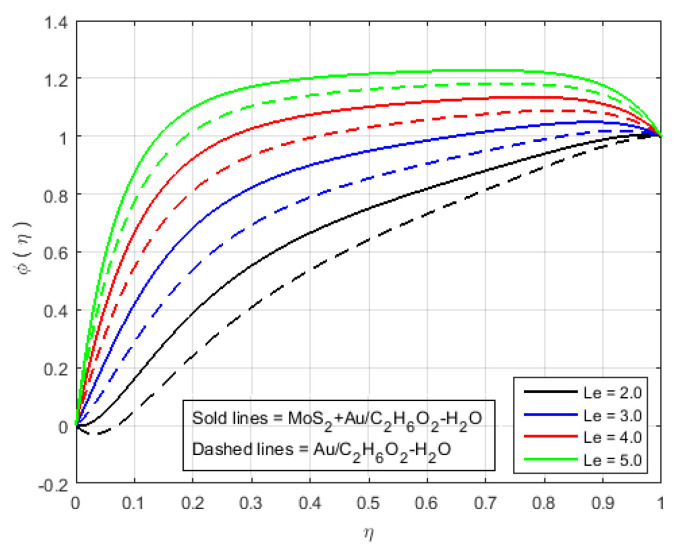
Impact of ϕ(η) for Le and fixed values of Sq=5.0, S=4.0, M=3.0, B=10.0, K=2.4, Q=0.6, $K_{1}^{*}=1.3,\;$Sc=0.7, Pr=6.2, Nt=1.4, Nb=1.5, Fr=1.5, $\Phi_{1}=\Phi_{2}=0.3$.

**Table 1 nanomaterials-12-00876-t001:** Thermophysical properties of nanoparticles (molybdenum di-sulfide, gold) and ethylene glycol with water at 20 °C.

Physical Properties	$\rho \left(\frac{\mathrm{kg}}{m^{3}}\right)$	$c_{p}\left(\frac{J}{\mathrm{kgK}}\right)$	$\kappa \left(\frac{W}{\mathrm{mK}}\right)$	$\sigma \left(\mathrm{sm}{}^{-1}\right)$
MoS${}_{2}$	5060	397.21	904.4	$17.9\times 10^{6}$
Au (Gold)	19300	130	310	$4.5\times 10^{7}$
$C_{2}H_{6}O_{2}-H_{2}O$	1063.8	3630	0.387	0.00509

**Table 2 nanomaterials-12-00876-t002:** Comparison of the numerical results of $f^{\prime \prime }\left(1\right)$ for the upper plate and $f^{\prime \prime }\left(0\right)$ for the lower plate when λ = 1, Fr = $K_{1}^{*}$ = *B* = Sq = 0, and $\Phi_{1}$ = $\Phi_{2}$ = 0, with different values of *M* and *S*.

			$f^{\prime \prime }\left(1\right)$		$f^{\prime \prime }\left(0\right)$
M	S	**Present**	**Najiyah S.K. et al. [28]**	**Present**	**Najiyah S.K. et al. [28]**
0	0.5	4.7133540	4.7133028	−7.4111453	−7.4111525
1	0.5	4.7390482	4.7390165	−7.5916641	−7.5916177
4	0.5	4.8202618	4.8202511	−8.1103709	−8.1103342
9	0.5	4.3964712	4.9648698	−8.9100425	−8.9100956
4	0.0	1.8424315	1.8424469	−4.5878710	−4.5878911
4	0.3	3.6536010	3.6536948	−6.6656598	−6.6656620
4	0.6	5.3912487	5.3912475	−8.8514001	−8.8514442
4	1.0	7.5934008	7.5934262	−11.9485561	−11.9485843

**Table 3 nanomaterials-12-00876-t003:** Comparison of the numerical results of $f^{\prime \prime }\left(1\right)$ for the upper plate when λ = 1, Fr = $K_{1}^{*}$ = *B* = *K* = 0, Q=Pr=Nb=Nt=Sc=Le=δ=ω = 0, and $\Phi_{1}$ = $\Phi_{2}$ = 0, with different values of *M*, *Sq*, and *S*.

				$f^{\prime \prime }\left(1\right)$
M	Sq	S	Present	Najiyah S.K. et al. [28]
0	1	0.5	1.814699	1.814634
0.25	1	0	−1.171512	−1.171551
0.25	1	0.5	1.808145	1.808177
0.25	0	0.5	4.719601	4.719656
0.25	1.5	0.5	0.283965	0.283948
0.25	1	1	4.573012	4.573016
1	1	0.5	1.789303	1.789372

**Table 4 nanomaterials-12-00876-t004:** Comparison of the numerical and analytical results of three methods PCM, BVP4C, and HAM for the skin friction and Nusselt number, with various physical parameters and $\Phi_{1}$ = $\Phi_{2}$ = 0.1.

	PCM	BVP4C	HAM	PCM	BVP4C	HAM
Sq	$f^{\prime \prime }\left(1\right)$	$f^{\prime \prime }\left(1\right)$	$f^{\prime \prime }\left(1\right)$	$-\theta^{\prime }\left(1\right)$	$-\theta^{\prime }\left(1\right)$	$-\theta^{\prime }\left(1\right)$
0	3.0934	3.0931	3.0926	−0.8443	−0.8440	−0.8438
0.1	3.0988	3.0981	3.0978	−0.8615	−0.8612	−0.8608
0.2	3.1042	3.1038	3.1033	−0.8791	−0.8788	−0.8783
0.3	3.1096	3.1091	3.1087	−0.8969	−0.8963	−0.8962
0.4	3.1151	3.1145	3.1139	−0.9151	−0.9146	−0.9140
0.5	3.1206	3.1201	3.1210	−0.9336	−0.9331	−0.9325
0.6	3.1261	3.1255	3.1250	−0.9523	−0.9518	−0.9511
0.7	3.1316	3.1310	3.1303	−0.9714	−0.9709	−0.9704
0.8	3.1372	3.1365	3.1360	−0.9908	−0.9902	−0.9914
0.9	3.1428	3.1423	3.1416	−1.0105	−1.0100	−1.0111
1	3.1484	3.1478	3.1471	−1.0306	−1.0302	−1.0310

## Data Availability

The data is available on reasonable request from the corresponding author.

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
