# Peer review of "Numerical Simulation of a Time-Dependent Electroviscous and Hybrid Nanofluid with Darcy-Forchheimer Effect between Squeezing Plates"

_nanomaterials, 2022, doi:10.3390/nano12050876_

Round 1

Reviewer 1 Report

Comments on “Numerical Simulation of a Time-dependent Electro-viscous and Hybrid Nanouid with Darcy-Forchheimer Eect between Squeezing Plates”

1- Check the English and make sure typos and grammatical errors are addressed.

2- Introduction should review state-of-the-art on heat transfer and advanced thermal engineering in nano-suspensions. For example, searching the literature, following papers are highly suggested to be read and used: Micropolar fluid past a convectively heated surface embedded with nth order chemical reaction and heat source/sink.  Evaluating the unsteady casson nanofluid over a stretching sheet with solar thermal radiation: An optimal case study.  Numerical analysis of dual variable of conductivity in bioconvection flow of Carreau–Yasuda nanofluid containing gyrotactic motile microorganisms over a porous medium.

3- Add more quantitative data to the abstract and conclusion to reflect the findings and main contribution of this work.

4- Why authors chose MoS2?Au=C2H6O2?H2O) as hybrid nanofluids? What is the advantage of these hybrids over other nanofluids?

5-What does h(t) and Vh physically mean? Some non-dimensional parameters must be defined based on the physics of the problem.

Overall, the paper can be accepted after addressing the above comments.

Author Response

TO

Editor-in-Chief
Nanomaterials

Manuscript ID: Nanomaterials-1542756                                                                                                                                                                                                                07 February 2022

Review submission regarding paper title:

Numerical Simulation of a Time-dependent Electro-Viscous and Hybrid Nanofluid with Darcy-Forchheimer Effect between Squeezing Plates

Authors: Muhammad Sohail Khan, Sun Mei, Shabnam, Unai Fernandez-Gamiz, Samad Noeiaghdam, Aamir Khan

  1. School of Mathematical Sciences, Jiangsu University, Zhenjiang 212013, Jiangsu, China.
  2.  Nuclear Engineering and Fluid Mechanics Department, University of the Basque Country UPV/EHU, Nieves Cano 12, 01006 Vitoria-Gasteiz, Spain.
  3. Industrial Mathematics Laboratory, Baikal School of BRICS, Irkutsk National Research Technical University, Irkutsk, 664074, Russia.
  4.  Department of Applied Mathematics and Programming, South Ural State University, Lenin prospect 76, Chelyabinsk, 454080, Russia.
  5. Department of Pure and Applied Mathematics, University of Haripur, 22620, KPK, Pakistan.

************************************************************************

REPLY TO REVIEWER 1

Comments on “Numerical Simulation of a Time-dependent Electro-viscous and Hybrid Nanouid with Darcy-ForchheimerEect between Squeezing Plates”

Q1.Check the English and make sure typos and grammatical errors are addressed.

Ans. English Language and typos errors have been corrected to the best of our knowledge.

Q2. Introduction should review state-of-the-art on heat transfer and advanced thermal engineering in nano-suspensions. For example, searching the literature, following papers are highly suggested to be read and used: Micropolar fluid past a convectively heated surface embedded with nth order chemical reaction and heat source/sink.  Evaluating the unsteady casson nanofluid over a stretching sheet with solar thermal radiation: An optimal case study.  Numerical analysis of dual variable of conductivity in bioconvection flow of Carreau–Yasuda nanofluid containing gyrotactic motile microorganisms over a porous medium.

Ans. All the suggested papers have been cited in the introduction section.  (See page 4, 5)

Q3. Add more quantitative data to the abstract and conclusion to reflect the findings and main contribution of this work.

Ans. The abstract and conclusion sections have been revised.

Q4. Why authors chose MoS2?Au=C2H6O2?H2O) as hybrid nanofluids? What is the advantage of these hybrids over other nanofluids?

Ans. These hybrid particles have been chosen because of their heat transfer properties in the ethylene-glycol + water. These particles improve the heat transfer rate and this was one of the tasks of this research work.

Q5. What does h(t) and Vh physically mean? Some non-dimensional parameters must be defined based on the physics of the problem.

Ans. h(t) represent the distance between the two plates as shown in figure 1. Vh has been deleted.

Overall, the paper can be accepted after addressing the above comments.

Thank you for the nice comments and for accepting this research work.

Reviewer 2 Report

Please see file attached.

Author Response

TO

Editor-in-Chief
Nanomaterials

Manuscript ID: Nanomaterials-1542756                                                                                                                                                                                                                07 February 2022

Review submission regarding paper title:

Numerical Simulation of a Time-dependent Electro-Viscous and Hybrid Nanofluid with Darcy-Forchheimer Effect between Squeezing Plates

Authors: Muhammad Sohail Khan, Sun Mei, Shabnam, Unai Fernandez-Gamiz, Samad Noeiaghdam, Aamir Khan

  1. School of Mathematical Sciences, Jiangsu University, Zhenjiang 212013, Jiangsu, China.
  2.  Nuclear Engineering and Fluid Mechanics Department, University of the Basque Country UPV/EHU, Nieves Cano 12, 01006 Vitoria-Gasteiz, Spain.
  3. Industrial Mathematics Laboratory, Baikal School of BRICS, Irkutsk National Research Technical University, Irkutsk, 664074, Russia.
  4.  Department of Applied Mathematics and Programming, South Ural State University, Lenin prospect 76, Chelyabinsk, 454080, Russia.
  5. Department of Pure and Applied Mathematics, University of Haripur, 22620, KPK, Pakistan.

 ************************************************************************

REPLY TO REVIEWER 2

Q1. Many typographic, syntactic, and orthographic errors

REPLY: All the grammatical and typo errors have been corrected to the best of our knowledge.

Q2. Repeated text: micro-electromechanical systems, etc.

REPLY: The sentence has been corrected. (See with colored red on page 1).

Q3. Is there no connection/dependence between l, vf, and b in p.5?

REPLY: The parameter l is used mistakenly. The parameter is now rechecked and defined in Figure 1, vf is the kinematic viscosity of the base fluid and b is the constant used for the representation of the base fluid.

Q4. P.4 states that “Single -phase mathematical model of Au-(MoS2)/water is formulated” however ethylene glycol is also included in the base fluid.

REPLY: The mistake has been corrected. The base fluid is ethylene-glycol + water. This sentence is rewritten and correct in Abstract and also on page.4.

Q5. p.7: S = V0/hb, what is h, is it not a function of t? Is it l, i.e. h at t = 0?

REPLY:  Yes, It is l (See page 7)

Q6. Figs 2,3,4,10: rectangular inlets are zooms at different positions of the graphs than those shown, and as such they are misleading.

REPLY: These graphs are drawn again and the effect of the parameters is now visible. (See with colored red on pages 14-18)

Q7. P.12: “….these two types of behaviour are displayed in Figure.”Which one?

REPLY: These types of behavior mean one of becoming parabolic in the middle area and the second opposite behavior near the two plates.

Q8. C2H6O2-H2O and Table 1 properties: what is the effect of the organic substance addition on the water, and what is its proportion in the mixture? What are the units of Table 1 parameters, in S.I.?
o of Au is 19300 kg/m3, NOT 19.30 shown in Table 1!!
o Cp of Au is ~130 J/kg K, NOT 317 shown in Table 1!!
o of Au is ~310 W/m K, NOT 0.128 shown in Table 1!!

Are these value actually used?? How do they compare to MoS2 properties now?

REPLY: Table 1 has been corrected. The SI units are added with each physical property.

Q9. Eq (2): What is F0 ? What is K* ? Should it not be B instead of B0 in this eq.?

REPLY: K*  is the permeability of porous space, F0  =  F is the non-uniform inertia coefficient, B replace with H and B0 is the strength of magnetic field. These mistakes are corrected (See Page 5, 7).

Q10. There are two definitions of B, B0/(1-at) and k2 , should the 2nd one be for B0?

REPLY: These mistakes have been corrected. (H = B is the inclusion of time-dependent magnetic field, B0 is the strength of the magnetic field, and B = k2…is fixed at a constant temperature) (See Page 5, 7)

Q11. Eq (5): Should there be all pluses (+) instead of (-) before all the terms in the RHS parenthesis?

REPLY: Eq 5 has been rechecked and found correct. This equation can be verify from the reference paper [38, 39]

Q12. Eq (6): What is T0 ? Reference temp., initial temp., T1-T2 or (T1+T2)/2? What is Tm in Eq (7)?

REPLY: The boundary conditions are corrected. (See Page 6)

Q13. There are two definitions of U at last paragraph of p.5. Should there be (-) in the 2nd RHS term of the first definition (stream function)? Is the second definition (electrical potential) referring to W?

REPLY: These mistakes are corrected. (See Page 5,6)

Q14. Eq. (7): What is concentration C, what does it represent? No such description in formulation. Is it referring to n+ and n- species? Then, is it Nernst-Planck not sufficient? There is no mass transport eq. in Ref. 29 mentioned as source!

REPLY: C is used for the mass transfer equation and is mention as equation 7 in the formulation section.  (See Page 5)

Q15. Eq (8): What are MS and CNT subscripts standing for? Why both 1 and 2 are for hnf/ f correlations) which is stated as source! Same goes for ( Cp)hnf /( Cp)f , hnf/ f, khnf/kf, hnf/ f !! e.g., there is no kbf in Ref. 30, and no distinction between kf and kbf is given in the MS! Which physical models then do they represent? Are they Maxwell-based? Brinkman-based?

REPLY: All these mistakes have been corrected. See equation 8 for the updated values (See page 7).

Q16. Eqs (9): Should T=T1-T0 instead of Tl-T0 and C=C1-C0 instead of Cl-C0 at y=0 BC? Should T=T2-T0 instead of Tl-T0 (and for C) at y=h(t) BC? Is not -1, according to the uw velocity description at p.5? >0 refers to the stretching lower plate have the value of 1 in Tables 2 & 3? What is its value in the rest of the results/figures?

REPLY: All the boundary conditions have been corrected.

Q17. Eqs (10): no y dependence of the correlation! Is something else than U ?

REPLY: The mistake has been corrected. Eta is a function of y and y has been added in the expression of eta.  (see page 6)

Q18. Eq (11): should it not be f '''' (=f iv , f (4)), instead of f ''', in the LHS? Also ( hnf/ f)/( hnf/ f) should be in front of all RHS terms, not for the given parenthesis only! Also subscript ( )hnf should be used instead of ( )nf terms in all Eqs (11)-(16) ! And what is the definition of Fr (inertia coefficient) used there? What is R in the last term?

REPLY: Equation 11 has been corrected. F’’’ has been converted to f’’’’. Also, nf has been converted to hnf and hf to bf.  (see pages 6- 7)

Q19. Eqs (12-14) what is Delta1?

REPLY: The value of Delta = x^2/l^2 has been added. (see page 7)

Q20. Eq. (17) what is w? If it is not zero, why Figs 8b, 9a-b & 10 start from zero at =0

REPLY: w = C1-C0 / C2- C0 and its value is 0.

Q21. P.8 section: Introducing parameter p gives ODEs in the p-parameter group p q for instance in Eq. 28, the P4' description, why P4 is replaced by (P4-1)q in the P3P4 term and not in the P4 term? Same goes for P8', P10'.

REPLY: These mistakes are corrected and replaced p = q.

Q22. Table 2 & 3: What are the values of all other dim/less numbers, besides , Sq, S, M, not mentioned used for the validation with Ref. 30, are they all zero? Unity? Since the base fluids and the nanoparticles used in the present work are substantially different from the ones used in Ref. 30, how can the authors perform a direct comparison?

REPLY: Yes, the nanoparticles in the two papers are different but in both papers the results match because in both papers the nanoparticles are taken zero. Also, in our problem the new parameters are taken zero for the sake for comparison of the results.

Q23. Results and Discussion section is not coherent and comprehendible as it seems hastily written, paragraphs are very large with no clear distinction between cases/figures description.

REPLY: The result and discussion section has been modified. All the grammatical and typo errors have been also corrected.

Q24. p.12: Influence of Sq at f( ) has been displayed graphically in both cases of MoS2-Au/C2H6O2-H2O and MoS2-C2H6O2-H2O is achieved for MoS2-C2H6O2-H2O and MoS2-Au/C2H6O2-H2O for MoS2-Au/C2H6O2-H2O and MoS2-C2H6O2-H2O Au-C2H6O2-H2O, not MoS2-Au/C2H6O2-H2O, is mentioned in the legends there (and other figures)! And if so, what are the values of 1 &2 in all these figures/results?

REPLY: All the figures are drawn again and the values of phi1 and phi2 are now mentioned in the captions of each figure.

Q25. p Nt and Nb at the concentration and temperature profile are illustrated through Figure 9. it has been noticed that the temperature distribution reduces monotonically from = 0 to No ( ) profile on Fig.9 ! No such monotonic decline can be observed on any Figure!!

REPLY: The discussion on figure 9 has been revised.

Q26. Fig.4: what is the value of Fr on (a)? of M on (b)?

REPLY: The value of Fr in figure 4 has been added.

Q27. Fig.3: S variation is mentioned in the caption, Sq is shown in the legends and described in text, which one is varied and which one is constant (to what value)?

REPLY: S is written mistakenly. The squeezing parameter is represented by Sq. The mistake has been corrected.

Round 2

Reviewer 1 Report

Accept

Author Response

All the grammatical and typo errors have been corrected to the best of our knowledge . Thanks

Reviewer 2 Report

Please see file attached.

Author Response

TO

Editor-in-Chief
Nanomaterials

Manuscript ID: Nanomaterials-1542756                                                                                                                                                                                                                19 February 2022

Review submission regarding paper title:

Numerical Simulation of a Time-dependent Electro-Viscous and Hybrid Nanofluid with Darcy-Forchheimer Effect between Squeezing Plates

Authors: Muhammad Sohail Khan, Sun Mei, Shabnam, Unai Fernandez-Gamiz, Samad Noeiaghdam, Aamir Khan

  1. School of Mathematical Sciences, Jiangsu University, Zhenjiang 212013, Jiangsu, China.
  2.  Nuclear Engineering and Fluid Mechanics Department, University of the Basque Country UPV/EHU, Nieves Cano 12, 01006 Vitoria-Gasteiz, Spain.
  3. Industrial Mathematics Laboratory, Baikal School of BRICS, Irkutsk National Research Technical University, Irkutsk, 664074, Russia.
  4.  Department of Applied Mathematics and Programming, South Ural State University, Lenin prospect 76, Chelyabinsk, 454080, Russia.
  5. Department of Pure and Applied Mathematics, University of Haripur, 22620, KPK, Pakistan.

 ************************************************************************

REPLY TO REVIEWER 2

Q1. Many typographic, syntactic, and orthographic errors

REPLY: All the grammatical and typo errors have been corrected to the best of our knowledge (See Page 1-3).

Q2. Repeated text: micro-electromechanical systems, etc.

REPLY: The sentence has been corrected. (See with colored red on page 1).

Q3. Is there no connection/dependence between l, vf, and b in p.5?

REPLY: The parameter l is used mistakenly. The parameter is now rechecked and defined in Figure 1, vf is the kinematic viscosity of the base fluid and b is the constant used for the representation of the base fluid.

Q4. P.4 states that “Single -phase mathematical model of Au-(MoS2)/water is formulated” however ethylene glycol is also included in the base fluid.

REPLY: The mistake has been corrected. The base fluid is ethylene-glycol + water. This sentence is rewritten and correct in Abstract and also on page.4.

Q5. p.7: S = V0/hb, what is h, is it not a function of t? Is it l, i.e. h at t = 0?

REPLY:  Yes, It is l (See page 7)

Q6. Figs 2,3,4,10: rectangular inlets are zooms at different positions of the graphs than those shown, and as such they are misleading.

REPLY: These graphs are drawn again and the effect of the parameters is now visible. (See with colored red on pages 14-18)

Q7. P.12: “….these two types of behaviour are displayed in Figure.”Which one?

REPLY: The sentences has been corrected. (See page 13-14)

Q8. C2H6O2-H2O and Table 1 properties: what is the effect of the organic substance addition on the water, and what is its proportion in the mixture? What are the units of Table 1 parameters, in S.I.?
o of Au is 19300 kg/m3, NOT 19.30 shown in Table 1!!
o Cp of Au is ~130 J/kg K, NOT 317 shown in Table 1!!
o of Au is ~310 W/m K, NOT 0.128 shown in Table 1!!

Are these value actually used?? How do they compare to MoS2 properties now?

REPLY: Table 1 has been corrected. (See page 11)

Q9. Eq (2): What is F0 ? What is K* ? Should it not be B instead of B0 in this eq.?

REPLY: K*  is the permeability of porous space, F0  =  F is the non-uniform inertia coefficient, B replace with H and B0 is the strength of magnetic field. These mistakes are corrected (See Page 5, 7).

Q10. There are two definitions of B, B0/(1-at) and k2 , should the 2nd one be for B0?

REPLY: These mistakes have been corrected. (H = B is the inclusion of time-dependent magnetic field, B0 is the strength of the magnetic field, and B = k2…is fixed at a constant temperature) (See Page 5, 7)

Q11. Eq (5): Should there be all pluses (+) instead of (-) before all the terms in the RHS parenthesis?

REPLY: Eq 5 has been rechecked and found correct. This equation can be verify from the reference paper [38, 39]

Q12. Eq (6): What is T0 ? Reference temp., initial temp., T1-T2 or (T1+T2)/2? What is Tm in Eq (7)?

REPLY: Definition has been found and corrected. (See Page 7, 8)

Q13. There are two definitions of U at last paragraph of p.5. Should there be (-) in the 2nd RHS term of the first definition (stream function)? Is the second definition (electrical potential) referring to W?

REPLY: These mistakes are corrected. Fr denote for Forchheimer number. (See Page 6, 7)

Q14. Eq. (7): What is concentration C, what does it represent? No such description in formulation. Is it referring to n+ and n- species? Then, is it Nernst-Planck not sufficient? There is no mass transport eq. in Ref. 29 mentioned as source!

REPLY: C is used for the mass transfer equation and is mention as equation 7 in the formulation section.  (See Page 5)

Q15. Eq (8): What are MS and CNT subscripts standing for? Why both 1 and 2 are for hnf/ f correlations) which is stated as source! Same goes for ( Cp)hnf /( Cp)f , hnf/ f, khnf/kf, hnf/ f !! e.g., there is no kbf in Ref. 30, and no distinction between kf and kbf is given in the MS! Which physical models then do they represent? Are they Maxwell-based? Brinkman-based?

REPLY: All these mistakes have been corrected. See equation 8 for the updated values (See page 6).

Q16. Eqs (9): Should T=T1-T0 instead of Tl-T0 and C=C1-C0 instead of Cl-C0 at y=0 BC? Should T=T2-T0 instead of Tl-T0 (and for C) at y=h(t) BC? Is not -1, according to the uw velocity description at p.5? >0 refers to the stretching lower plate have the value of 1 in Tables 2 & 3? What is its value in the rest of the results/figures?

REPLY: All the boundary conditions have been corrected.

Q17. Eqs (10): no y dependence of the correlation! Is something else than U ?

REPLY: The mistake has been corrected. Eta is a function of y and y has been added in the expression of eta.  (see page 6)

Q18. Eq (11): should it not be f '''' (=f iv , f (4)), instead of f ''', in the LHS? Also ( hnf/ f)/( hnf/ f) should be in front of all RHS terms, not for the given parenthesis only! Also subscript ( )hnf should be used instead of ( )nf terms in all Eqs (11)-(16) ! And what is the definition of Fr (inertia coefficient) used there? What is R in the last term?

REPLY: These mistakes are corrected.  (see pages 6- 9)

Q19. Eqs (12-14) what is Delta1?

REPLY: The value of Delta = alpha^2/l^2 has been corrected. (see page 7)

Q20. Eq. (17) what is w? If it is not zero, why Figs 8b, 9a-b & 10 start from zero at =0

REPLY: Found and correct [28]. (See page 8)

Q21. P.8 section: Introducing parameter p gives ODEs in the p-parameter group p q for instance in Eq. 28, the P4' description, why P4 is replaced by (P4-1)q in the P3P4 term and not in the P4 term? Same goes for P8', P10'.

REPLY: These mistakes are corrected. (See page 9)

Q22. Table 2 & 3: What are the values of all other dim/less numbers, besides , Sq, S, M, not mentioned used for the validation with Ref. 30, are they all zero? Unity? Since the base fluids and the nanoparticles used in the present work are substantially different from the ones used in Ref. 30, how can the authors perform a direct comparison?

REPLY: The mistake is corrected and the parameters values are mentioned. (See page 11)

Q23. Results and Discussion section is not coherent and comprehendible as it seems hastily written, paragraphs are very large with no clear distinction between cases/figures description.

REPLY: The result and discussion section has been modified. (See page 13, 14)

Q24. p.12: Influence of Sq at f( ) has been displayed graphically in both cases of MoS2-Au/C2H6O2-H2O and MoS2-C2H6O2-H2O is achieved for MoS2-C2H6O2-H2O and MoS2-Au/C2H6O2-H2O for MoS2-Au/C2H6O2-H2O and MoS2-C2H6O2-H2O Au-C2H6O2-H2O, not MoS2-Au/C2H6O2-H2O, is mentioned in the legends there (and other figures)! And if so, what are the values of 1 &2 in all these figures/results?

REPLY: All the figures are corrected and modified.

Q25. p Nt and Nb at the concentration and temperature profile are illustrated through Figure 9. it has been noticed that the temperature distribution reduces monotonically from = 0 to No ( ) profile on Fig.9 ! No such monotonic decline can be observed on any Figure!!

REPLY: The discussion and figures has been revised. (See page 14, 15)

Q26. Fig.4: what is the value of Fr on (a)? of M on (b)?

REPLY: The value has been mentioned. (See page 16)

Q27. Fig.3: S variation is mentioned in the caption, Sq is shown in the legends and described in text, which one is varied and which one is constant (to what value)?

REPLY: S is written mistakenly. The squeezing parameter is represented by Sq. The mistake has been corrected.
